# Defect Detection and Depth Estimation in Composite Materials for Pulsed Thermography Images by Nonuniform Heating Correction and Oriented Gradient Information

**DOI:** 10.3390/ma16082998

**Published:** 2023-04-10

**Authors:** Jorge Erazo-Aux, Humberto Loaiza-Correa, Andrés David Restrepo-Girón, Clemente Ibarra-Castanedo, Xavier Maldague

**Affiliations:** 1Escuela de Ingeniería Eléctrica y Electrónica, Universidad del Valle, Cali 760032, VA, Colombia; humberto.loaiza@correounivalle.edu.co (H.L.-C.); andres.david.restrepo@correounivalle.edu.co (A.D.R.-G.); 2Facultad de Ingeniería, Institución Universitaria Antonio José Camacho, Cali 760046, VA, Colombia; 3Computer Vision and Systems Laboratory, Laval University, Quebec City, QC G1V 0A6, Canada; clemente.ibarra-castanedo@gel.ulaval.ca (C.I.-C.); xavier.maldague@gel.ulaval.ca (X.M.)

**Keywords:** pulsed thermography, composite materials, automated defect detection, estimation of depth, contrast enhancement, histograms of oriented gradients

## Abstract

Pulsed thermography is a nondestructive method commonly used to explore anomalies in composite materials. This paper presents a procedure for the automated detection of defects in thermal images of composite materials obtained with pulsed thermography experiments. The proposed methodology is simple and novel as it is reliable in low-contrast and nonuniform heating conditions and does not require data preprocessing. Nonuniform heating correction and the gradient direction information combined with a local and global segmentation phase are used to analyze carbon fiber-reinforced plastic (CFRP) thermal images with Teflon inserts with different length/depth ratios. Additionally, a comparison between the actual depths and estimated depths of detected defects is performed. The performance of the nonuniform heating correction proposed method is superior to that obtained on the same CFRP sample analyzed with a deep learning algorithm and the background thermal compensation by filtering strategy.

## 1. Introduction

Composite materials are currently of great interest for various industries since they are resistant to corrosion and fatigue and are stiffer and lighter than traditional materials, such as steel [1]. High-performance applications in the aerospace, naval, automotive, structural health and biomedical industries, among others, demand high-quality composites [2]. Evaluating the quality of these materials and the structures built from them is a fundamental task in modern production processes. Nondestructive testing (NDT) techniques facilitate inspection and quality control processes while avoiding compromising the integrity of the objects of interest. Infrared thermography (IRT) is an attractive option for NDT as it is fast, safe, noninvasive, and contact-free [3,4].

In thermal imaging, locating possible anomalies (detection) and estimating their attributes (characterization), such as their shape, size, depth or other properties, are not simple tasks. In addition, undesirable effects such as low contrast, the presence of noise, and nonuniform heating further complicate these tasks [5,6,7,8]. Traditional techniques for thermal information processing have focused on enhancing image contrast, in turn allowing subsequent defect detection and/or characterization processes. However, sometimes, a processing method does not have a clearly defined scope and does not present objective criteria for quantifying its performance. Various processing methods, for example, normalized contrast, traditional definitions of contrast [3], pulsed phase thermography (PPT) [9], background thermal compensation by filtering (BTCF) [10], differential absolute contrast (DAC) [11], polynomial regressions [12], and thermographic signal reconstruction (TSR) [13], are based on the analysis of spatial data, thermal profiles in the temporal domain, phase profiles in the frequency domain, combined information in different domains, heat propagation models, or mathematical transformations. In addition, according to the references consulted, few procedures have addressed the task of automatically detecting defects in thermal images as a preliminary step in characterization processes [14,15,16,17,18]. Using independent components, Rengifo et al. [14] analyzed complete thermal sequences to synthesize an image highlighting the inspected material anomalies. Florez-Ospina et al. [15,16], based on signal-to-noise ratio (SNR) maps derived from processed IR images, proposed a general framework for the semi-automatic calculation of segmentation algorithm parameters. Aguilera et al. [17] used a scale-invariant-feature-transform (SIFT)-like scale representation and key points (region of interest) to register far infrared and visible images. FInally, Maldague et al. [18] proposed an algorithm for subsurface defect extraction in infrared images.

Moreover, possibly the most important source of degradation caused by low contrast in thermal images derives from nonuniform background heating [19]. Related to the considerations described above, recent studies [10,12,20] have proposed decoupling background and defect information in thermograms, contrary to conventional contrast enhancement techniques that generally require defining healthy reference regions. Thus, a higher contrast thermogram is achieved by separating the background from the image and subtracting it from the original image. In this regard, in [12], the thermal behavior of the defect-free material (background) was obtained from a sixth-order polynomial regression. However, the defective regions of the material must be identified and eliminated first in order to reduce the distortion in the regression. Another alternative, presented in [10,20], proposes the use of two-dimensional median and Gaussian smoothing filters to suppress the defective regions of the material and obtain a smoothed image representing the background; however, the filters require parameters to be defined for their implementation. Most recent works reported in the literature use machine learning [21], deep learning models [22,23], convolutional neuronal networks [24], and actual datasets [25] or synthetic datasets [21] to train and validate the proposed approaches. Regarding the methods mentioned above, the reported methods must establish reference regions that suppose prior sample knowledge, apply smoothing filters and define their parameters, perform high-order polynomial regression, or make a complete or partial analysis of the temporal/frequency evolution of the temperature profile or its characteristics. This, together with the high volume of information in a sequence of thermal images, the low contrast, and the presence of several adverse effects, mandate advances in the defect detection process.

Based on our previous work [26,27], this paper proposes a procedure for automated defect detection in inspected composites using pulsed thermography. The main contributions of our approach compared to the already known works are that our development methodologies do not require adjusting parameters or selecting reference regions and do not require prior knowledge about the state of the sample to be analyzed. Additionally, a complete or partial analysis of the temporal evolution of the temperature or its characteristics is not necessary. The procedure consists of two methodologies: (1) the first automatically calculates the optimal parameters of a function that allows modeling the background temperature distribution of the image in order to improve its contrast and subsequently identify defects by a local segmentation technique, and (2) the second transforms the thermal image into gradient magnitude and angle information and builds directional histograms that, together with a simple statistical strategy, allow the identification of the regions containing heating patterns of defective and sound zones. The procedure was tested on synthesis thermal images, with each image representing the maxigram or discrete integral (sum) of an actual or simulated thermal sequence of a CFRP sample. Each sample contained 25 and 8 Teflon inserts with length/depth ratios between 15 and 75 and between 1.7 and 90, respectively. The performance of the procedure was evaluated for actual images using the precision-recall ratio in the F-indicator and using the area under the curve (AUC) indicator for simulated images. In maxigrams of actual sequences, F values between 0.60 and 0.67 were obtained. The average AUC performance values were (i) 0.98 in integral images and (ii) 0.89 in simulated sequence maxigrams. Additionally, a comparison between actual depths and estimated depths of detected defects was performed, and average percentage errors were found to be (i) 4.33% in simulated sequence maxigrams and (ii) 22.26% in actual sequence maxigrams. Finally, the estimated depth results were statistically validated using an analysis of variance (ANOVA).

This paper is organized as follows. Section 2 provides a detailed description of the proposed procedure and specifies the properties of the materials and the parameters fitted for the pulsed thermography experiment. Section 3 presents the results obtained and analyzed using the precision-recall ratio and the AUC indicator. These results are statistically validated with an analysis of variance. Additonally, the percentage error calculated between the actual and estimated depth of the detected defects are presented. Finally, Section 4 sets out the conclusion and proposes future directions.

## 2. Materials and Methods

Figure 1 shows a flowchart of the procedure proposed in this paper. It is necessary to define the presence or absence of noise in the thermal sequence based on the criteria of a human inspector. The upper part shows the flow and type of thermal information (image from a sequence or image synthesized from a sequence of images) entering the detection stage based on HOG. Similar information for the contrast enhancement-based detection stage is presented in the lower part of the figure. A detailed description of how each methodology works is developed in the following sections.

### 2.1. HOG for Defect Detection

In the thermograms of the inspected material, the segmentation of defective (ROI-d) and sound (ROI-s) regions of interest is performed in three stages as proposed in [26]. In the first stage, the gradient magnitude and angle information are calculated from the thermal image. Then, the gradient information is organized into histograms of oriented gradients, which represent local signatures of gradient orientation. Finally, with the signatures provided by these histograms, together with median-based image thresholding, the gradients corresponding to ROI-d and ROI-s are differentiated.

In the first stage, the magnitude (|∇I|=g(x,y)) and angle (θ) of the gradient are obtained for each image I(x,y) composing the thermal image sequence, as defined in Equation (Equation 1).
(1)$$\begin{array}{c} g\left(x,y\right)=\left|\nabla I\right|=\sqrt[{}]{{\frac{\partial I}{\partial x}}^{2}+{\frac{\partial I}{\partial y}}^{2}},\;\theta =\tan^{-1}\left(\frac{\partial I/\partial y}{\partial I/\partial x}\right). \end{array}$$

In the second stage, gradient profiles are constructed over small areas of each image to characterize the thermal changes as a function of the level of variation, direction, and relationships with neighboring areas. This procedure involves two main tasks: (1) construction of local histograms of gradient directions and (2) block normalization of the histograms. Equation (Equation 2) presents a general expression for obtaining the local histograms ($h_{i}$) in relation to the image subregions called cells ($C_{i}$) and blocks ($B_{j}$), the number of blocks (η) into which the image is divided, and the number of cells (γ) contained in each block. The working range is chosen between two interval options, (0°, 360°) or (0°, 180°), depending on how the oriented gradient information is organized. The division number (β) in the working range is defined to complete the construction of the HOG. Finally, the w parameter could be 1 if the analyzed region is contained in the work division of the histogram or zero otherwise, respectively
(2)$$\begin{array}{c} h_{C_{i}}^{B_{j}}\left(k\right)=\sum_{\left(x,y\right)\in C_{i}}w\cdot g\left(x,y\right)\;\left\{\begin{array}{c} \begin{array}{cc} \forall \;B_{j}\to j & =1,\ldots ,\eta \\ \forall \;C_{i}\in B_{j}\to i & =1,\ldots ,\gamma \\ k & =1,\ldots ,\beta \end{array} \end{array}\right. \end{array}.$$

Each block is divided into γ cells; then, the directional histogram is calculated in each cell, and a vector is subsequently generated by concatenating all the histograms of the cells. From the division of each element by the norm of this vector $\left|\right|H^{B_{j}}\left|\right|$, the normalization of all the elements of the histograms is obtained according to Equation (Equation 3).
(3)$$H^{B_{j}}=\left[h_{C_{1}}^{B_{j}},\ldots ,h_{C_{i}}^{B_{j}}\right],\;\to \;H_{norm}^{B_{j}}=H^{B_{j}}/\left|\right|H^{B_{j}}\left|\right|.$$

In this paper, the statistical median is used on the normalized histograms corresponding to each cell (${\tilde{h}}_{C_{i}}$) because this is a special case of an average limited to one or two values; it is a measure of central tendency of information that, in addition, reduces the effect of possible extreme values [28]. A matrix is constructed from the medians of these histograms. The element accumulating the highest number of observations (statistical frequency) in the matrix determines the threshold value (*th*) for automating the segmentation task defined in Equation (Equation 4), for which values less than or equal to *th* are associated with ROI-s, thus allowing their differentiation from ROI-d.
(4)$$I_{b}\left(x,y\right)=\left\{\begin{array}{cc} 1 & \mathrm{If}\;{\tilde{h}}_{C_{i}}>th\;\wedge \left(x,y\right)\in C_{i}\\ 0 & \mathrm{If}\;{\tilde{h}}_{C_{i}}\leq th\;\wedge \left(x,y\right)\in C_{i} \end{array}.\right.$$

### 2.2. Background Thermal Compensation by Parameter Optimization (BTCOp) with Gaussian Function Parameters

This procedure is described in detail in [27] and posits that the energy pulse used in a pulsed thermography (PT) experiment produces a thermal pattern in the inspected material that is related to the nonuniform heating effect. The pattern is exhibited by a temperature peak in the middle of the image that gradually subsides towards the edges. Previously, the nonuniform heating phenomenon has been approximated using bi-hexic functions [12] or using median-type [10] and Gaussian-type [20] filters. However, as in the case of many signals [3] and considering the physical properties of the sheets and the mechanism by which cooling affects them, the spatial distribution of heat could be considered a Gaussian-type process. Thus, this processing block allows for modeling the spatial behavior of the background thermal pattern as a two-dimensional Gaussian surface. The parameters of the two-dimensional function are calibrated from the experimental data using the least squares method.

Equation (Equation 5) shows an exponential representation of the Gaussian objective function (ϕ) describing the system related to the *k*-th image of a thermal sequence, and the vector x represents the spatial coordinates (*x*, *y*) within the image ($x\in N^{2}$, $x_{1}=1,2,\ldots ,m$ and $x_{2}=1,2,\ldots ,n$, where *m* and *n* are the numbers of rows and columns, respectively, of pixels in the image). The vector $\theta ={\left[A,\mu ,\Sigma \right]}^{T}$ contains the unknown parameters of this function to be estimated. The maximum magnitude of the Gaussian function is defined by the constant *A*${\left(\sqrt{2\pi |\Sigma |}\right)}^{-1}$, μ is the vector of means, and Σ is the covariance matrix [29].
(5)$$\phi^{k}\left(x,\theta \right)=e^{A}\cdot e^{-\frac{1}{2}\cdot {\left(x-\mu \right)}^{T}\Sigma^{-1}\left(x-\mu \right)},\;$$

If the difference between the value of the Gaussian objective (ϕ) and the value of a temperature sample (*T*) evaluated at the same spatial coordinate (x) is defined as an error (h), it is possible, using the natural logarithm, to construct a linear model of the *i-th* error term for each spatial location within an image (see Equation (Equation 6)).
(6)$$h_{i}\left(x_{\left(i\right)},\theta \right)=\ln\left(\phi \left(x_{\left(i\right)},\theta \right)\right)-\ln\left(T\left(x_{\left(i\right)}\right)\right).$$

This approach fits a least squares system and can be presented as an unconstrained optimization problem, as shown in Equation (Equation 7): (7)$$\begin{array}{cc} \min f(\theta ) & =\frac{1}{2}H^{T}\left(\theta \right)\cdot H\left(\theta \right)\\ \theta \in R^{5}, \end{array}$$
where the vector *H* contains a number of elements equal to the number of available measurements (*s*) in the thermal image (see Equation (Equation 8)).
(8)$$H\left(\theta \right)=\left[h_{1}\left(x_{\left(1\right)},\theta \right),\ldots ,h_{i}\left(x_{\left(i\right)},\theta \right),\ldots ,h_{s}\left(x_{\left(s\right)},\theta \right)\right].$$

The problem posed in Equation (Equation 7) can be solved using general optimization methods, with emphasis on gradient methods involving iterative processes. The detailed solution to the above problem is described in [27].

After obtaining the optimal parameter vector $\theta_{p}$, the parameter vector $\hat{\theta }$ in the original problem domain becomes available as a solution to the optimization problem by applying the exponential function to it. Evaluating $\hat{\theta }$ in Equation (Equation 5) yields $T_{B}^{k}$, which represents the estimated nonuniform background heating model for the *k-th* image (see Equation (Equation 9)).
(9)$$T_{B}^{k}=\phi^{k}\left(x,\hat{\theta^{k}}\right).$$

As proposed in [10], the background thermal compensation is achieved by subtracting from the original image $T^{k}$ the estimated model of the surface with nonuniform background heating $T_{B}^{k}$ (see Equation (Equation 10)).
(10)$$\Delta T_{BTCOp}^{k}=T^{k}-T_{B}^{k}.$$

Finally, it is possible to define a normalized version of Equation (Equation 10) using a reference image (see Equation (Equation 11)), which can be selected at the moment when the excess temperature is maximum or at the end of the thermal process [3,16].
(11)$$\Delta T_{BTCOpN}^{k}=\frac{T^{k}}{T^{ref}}-\frac{T_{B}^{k}}{T_{B}^{ref}}.\;$$

#### Defect Segmentation in Contrast-Enhanced Thermograms

At this stage, once the thermal images have been contrast-enhanced (see Section 2.2) a local median-based thresholding technique is applied to separate ROI-s and ROI-d. Similar to the HOG method (see Section 2.1), local structures called cells ($C_{i}$) are defined to calculate the median statistic using all elements contained in a region. The cell size for a region is defined heuristically by varying its value from the smallest to the largest size of the defects present in the inspected samples, with the latter value presenting the most favorable segmentation result. Once the cell sizes have been defined, the median value in each region is calculated and used as the local segmentation threshold (*th*${}_{i}$), which allows the automatic generation of a binary version $I_{b}\left(x,y\right)$ of the thermal image being analyzed (see Equation (Equation 12)).
(12)$$I_{b}\left(x,y\right)=\left\{\begin{array}{cc} 1 & \mathrm{If}\;\Delta T_{BTCOp}^{k}\left(x,y\right)>th_{i}\;\wedge \left(x,y\right)\in C_{i}\\ 0 & \mathrm{If}\;\Delta T_{BTCOp}^{k}\left(x,y\right)\leq th_{i}\;\wedge \left(x,y\right)\in C_{i}. \end{array}\right.$$

### 2.3. Description of the Set of Images

The images used in this work consisted of two sets of sequences. The first set was generated from an actual PT experiment, and the second set was generated by applying simulated PT experiments. These sets of sequences are similar to those used in [27] and supplemented in [30].

#### 2.3.1. Actual Images

Under the conditions described in Table 1, a PT experiment was performed, obtaining 300 actual 470×475 pixel images of a square carbon fiber-reinforced plastic (CFRP) sample with a lateral size of 300 mm, a thickness of 2 mm, a diffusivity (α) of $4.6\times 10^{-7}\frac{m^{2}}{s}$ and 25 Teflon inserts of different sizes and depths representing the defects (see Figure 2a).

#### 2.3.2. Simulated Images

The set of synthetic thermal images used in this work was synthesized using ThermoCalc-6L software. The simulated PT experiment was performed on an artificial CFRP specimen of planar geometry, square shape, lateral size of 200 mm, and thickness of 2 mm. The CFRP specimen contained simulated internal squared defects with fixed thickness and variable area and depth.

Using previous investigations [27,32,33] as references, Table 2 and Table 3 present the acquisition conditions adjusted in the simulation and the physical properties of the materials, respectively.

The different simulated thermal imaging sequences were obtained by varying the lateral size (**S**), thickness (**Th**), and depth (**D**) parameters of the defects (see Figure 2b). Adjusting each of these parameters produces a different sequence of synthetic images for each simulation, and a single value of S simultaneously adjusts the lateral size of all defects. The simulated data consist of 49 sequences of 1429 images each, for a total of 70,021 images of 200×200 pixels each. To evaluate the methodology proposed in this work, we used a set of 30 synthetic sequences from the total of 49 available in the complete set. The sequences chosen correspond to lateral size defects of 3 mm, 6 mm, 9 mm, 12 mm, and 15 mm. Thus, the simulated sequences encompass defect sizes similar to those found in the actual CFRP sample.

## 3. Results and Discussion

This section presents the results obtained for the proposed automated defect detection procedure and its subsequent depth estimation, applying different tests on the image sets described in Section 2.3. Sum-type [34] and maxigram [3] synthesis images were used for the tests. Sum-type images were chosen to evaluate the methodologies for defect detection on thermal information under low-contrast conditions. Maxigrams were chosen because they also allow estimating the depth of the detected anomalies. Although maxigrams can be obtained on image sequences without preprocessing, the temporal information concerning them is in the first instants of the sequence due to the energy generated by the PT experiment. This condition produces unreliable depth estimates with the chosen approach. Therefore, sum-type images were only used on raw images. Section 3.1 presents the detection performance for defects in simulated sum-type images. Section 3.2 shows performance results for maxigrams of simulated images, including depth estimation results for detected defects. Finally, Section 3.3 presents results for defect detection in actual thermal image maxigrams and depth estimation for the detected defects.

### 3.1. Defect Detection in Simulated Thermal Sequence Sum-Type Synthetic Images

Row 1 in Figure 3 shows sum-type synthetic images obtained for the CFRP material in simulated raw sequences. The CFRP sample contains internally defective regions of interest (ROI-d) with lateral sizes (**S**) of 3 mm (column 1), 6 mm (column 2), 9 mm (column 3), 12 mm (column 4), and 15 mm (column 5) located at a depth (**D**) of 1.0 mm. Row 2 shows the binary images resulting from the application of the HOG-based automated defect detection methodology [26].

In all the sum-type images in Figure 3 (row 1, columns 1 to 5) the effect of nonuniform heating can be seen, which manifests itself with a concentration of intensities greater in the center of the image and progressively decreasing towards the edges. This phenomenon, in addition to producing low contrast in the thermal images, prevents easy identification of the ROI-d, especially for smaller defects (e.g., columns 1 and 2). However, the HOG-based automated detection method shows acceptable results after evaluating different (**S**) conditions (see Figure 3, row 2). Supporting the results described above, the AUC performance values for depth (**D**) values of 0.1 mm, 0.4 mm, 0.7 mm, 1.0 mm, and 1.3 mm are given in Table 4. The AUC indicator is constructed by comparing the results obtained in the analyzed images with a template or binary reference image containing the correctly segmented defects. In our case, the AUC is used to evaluate the classification results (Equation (Equation 13)). An ideal system will achieve a true-positive rate (TPR) of 1.0 and a false-positive rate (FPR) of 0; thus, the ideal AUC will be 1.0. However, in practice, a reliable system should preferably have AUC values greater than 0.5.
(13)$$AUC=\frac{1}{2}\left(1+TPR-FPR\right)$$

### 3.2. Automated Defect Detection and Depth Estimation in Simulated Sequence Maxigrams

Row 1 in Figure 4 shows maxigrams of simulated CFRP sequences contrast-enhanced using the BTCOp method [27]. The CFRP sample contains ROI-d located at the same depth (**D**) and with lateral sizes (**S**) equal to those used in the test described in Section 3.1 (columns 1 to 5).

The improved information is seen with a dark and uniform intensity level that represents the background of the image, while the defective regions are easily distinguished with light intensity levels. Thus, the defects are observed for all the lateral size (**S**) conditions evaluated. In row 2 of the same figure, the binary images resulting from applying the methodology for automated defect detection based on the combined use of BTCOp and local thresholding described in Section 2.2 are shown. Table 5 presents the AUC performance indexes for the depth (**D**) and lateral size (**S**) values used with the simulated thermal sequence maxigrams. In general, the performance indexes are slightly lower than those obtained with the sum-type images processed with the HOG method.

With the HOG-based methodology, overly segmented ROI-d are obtained (see Figure 3, row 2). This result can be attributed to the geometry and resolution of the structures used to calculate the HOGs, as explained in [26]. On the other hand, after compensating in the images for nonuniform heating with the BTCOp method [27], more well-defined defects with more uniform sizes can be observed (see Figure 4, row 2). The BTCOp method uses all available samples in the image for estimating the background model parameters and therefore has no resolution constraints.
(14)$$h_{d}=A\cdot \sqrt{t_{C_{max}}}\cdot {\left(C_{max}\right)}^{n}.$$

Applying a linear regression method to Equation (Equation 14) [3], it is possible to calculate the coefficients *A* and *n* that allow automatic estimation of the depths ($h_{d}$) of the ROI-d detected. The values $C_{max}$ and $t_{Cmax}$ signify the maximum contrast value, in our case located at the element (pixel) having the maximum intensity value within the detected ROI-d, and its corresponding time, respectively.

With the estimated depth values (**D**${}_{e}=h_{d}$) for the ROI-d, percentage error values relative to the actual depths (**D**) of the defects are calculated (see Equation (Equation 15)). In a maxigram of simulated sequences, all ROI-d are at the same depth. Table 6 shows the calculated percentage errors for all evaluated length/depth ratios. No trends were identified in the results that relate the calculated error as a function of depth (**D**) or lateral size (**S**) of the defects. However, a maximum percentage error of 8.07% (**S** = 9 mm y **D** = 0.4 mm) and a minimum percentage error of 0.68% for (**S** = 3 mm y **D** = 0.1 mm) were observed.
(15)$$Error\left[\%\right]=\frac{\left|D-D_{e}\right|}{D}\cdot 100.\;$$

Figure 5 presents the results of the ANOVA that statistically complements the information described in Table 6. In this test, the null hypothesis ($H_{0}$) was that the mean values of estimated depth (**D**${}_{e}$) of the ROI-d are equal for the five lateral sizes (**S**) used. On the other hand, the alternative hypothesis ($H_{1}$) was that these mean depth values are different. Before performing this statistical test, the datasets were tested for normality.

The ANOVA test was repeated with the same conditions for each of the five depth (**D**) values evaluated (0.1 mm, 0.4 mm, 0.7 mm, 1.0 mm, and 1.3 mm). The datasets for each ANOVA corresponded to estimated depth (**D**${}_{e}$) values for the eight ROI-d present in each maxigram relative to the five lateral size (**S**) conditions (see Figure 5a–e). Thus, each (**D**) condition was represented by 39 degrees of freedom (eight **D**${}_{e}$ values for five lateral size values).

From the *p*-values obtained in the tests and with a certainty of 95%, it is possible to affirm that the defects with **S** values of 6 mm, 9 mm, 12 mm, and 15 mm did not differ significantly in **D**${}_{e}$. However, they exhibited higher **D**${}_{e}$ values than those found for a lateral size of 3 mm (see Table 6). These differences could be explained by noise levels and nonuniform heating correction affecting small defects more than large ones. The above behavior was maintained for all depth (**D**) values evaluated.

### 3.3. Automated Defect Detection and Depth Estimation in Actual Sequence Maxigrams

In this section, we present the detection of defective regions by the techniques proposed in our study (BTCOp and normalized BTCOp, BTCOpN) and using traditional thermal information processing techniques: normalized contrast (CN), DAC, BTCF, and normalized BTCF (BTCFN). For ROI-d segmentation with the traditional methods, the classical Canny edge detection algorithm was used. For the BTCOp and BTCOpN methods, the automated detection process proposed in this paper and described in Section 2.2 was used.

Figure 6 shows in the first row the sequence maxigrams of the actual CFRP sample (see Section 2.3.1) contrast-enhanced using the CN (column 1), DAC (column 2), BTCF (column 3), BTCFN (column 4), BTCOp (column 5), and BTCOpN (column 6) methods. In the second row are the corresponding binary images resulting from the detection process.

The binary images produced after analyzing the contrast-enhanced maxigrams show that the BTCOpN, BTCFN, and BTCOp methods correctly detected 19, 18, and 13 ROI-d and falsely detected 1, 1, and 5 ROI-d, respectively (Figure 6, row 2, columns 6, 4 and 3). The BTCF, CN, and DAC methods underperformed, correctly detecting 11, 11, and 5 ROI-d, respectively, and falsely detecting 2, 2, and 5 ROI-d, respectively (Figure 6, row 2, columns 3, 1, and 2).

With the normalized BTCOpN and BTCFN techniques, maxigrams with better contrast were obtained (in comparison with the maxigrams generated with the other methods, which were BTCF, BTCOp, CN, and DAC). The result described above is consistent with the number of correct detections obtained using these methods and additionally with the superior performance displayed by these techniques as a function of the signal-to-noise ratio (SNR) [27]. The performance of the BTCOpN method is similar to the result obtained for the same actual CFRP sample recently analyzed with a deep learning algorithm [23], where 17 correct detections and 4 false ROI-d detections were reported. Table 7 shows the performance indicators that were calculated using the different processing techniques for the maxigrams in Figure 6. Using the recall indicator, which represents the proportion of actual defects that were correctly identified, the best performance was exhibited by the BTCOpN technique (0.84), followed by the BTCFN (0.72), BTCOp (0.61), BTCF (0.56), CN (0.39), and DAC (0.18) methods.

In terms of accuracy, the BTCOpN and BTCOp methods showed values close to 0.56, representing a nonnegligible level of false detections. However, these were mostly located around the correctly detected ROI-d, which would facilitate the future implementation of a strategy to improve shape characterization in defects (ROI-d). This phenomenon, which can be interpreted as oversegmentation, can be attributed to the heat scattering effect that occurs at the boundaries between the sound and defective regions of the inspected material, the resolution that was defined for the regions, and the calculation of the local thresholds.

Following a procedure similar to that described in Section 3.2, Table 8 and Table 9 present the estimated depth (**D**${}_{e}$) values and the calculated percentage errors between the actual depths (**D**) of the defects and the estimated depths (**D**${}_{e}$) that were obtained using the BTCOp and BTCOpN contrast enhancement methods.

As with the simulated sequence maxigrams, no trends were identified in the results for the actual sequence maxigrams for the error values calculated as a function of defect depth (**D**) or lateral size (**S**). Using BTCOp for contrast enhancement, a maximum relative percentage error of 13.04% for **S** = 7 mm and a minimum relative percentage error of 0.39% for **S** = 5 mm were found. On the other hand, with the BTCOpN method, the maximum value was 161.26% for **S** = 3 mm, and the minimum was 0.11% for **S** = 15 mm.

Figure 7 presents ANOVA results to complement the statistical analysis of the information in Table 8 and Table 9. Similar to the hypotheses examined in Section 3.2, the null hypothesis (H${}_{0}$) was that the mean estimated depth values (**D**${}_{e}$) of the detected ROI-d are equal in relation to the actual depth values (**D**) (0.2 mm, 0.4 mm, 0.6 mm, 0.8 mm, and 1.0 mm). The alternative hypothesis (H${}_{1}$) was that these mean depth values are different.

Based on the *p*-values obtained in the tests and with a certainty of 95%, it is possible to state the following:On the maxigram of the actual sequence compensated with the BTCOp method, in the detected ROI-d and using the values of (**D**${}_{e}$), statistically significant differences were found for the four actual depths corresponding to the defects (see the left side of Figure 7). In this case, the numbers of defects that could be detected were five for **D** = 0.2 mm, four for **D** = 0.4 mm, two for **D** = 0.6 mm, and two for **D** = 0.8 mm.On the maxigram of the same actual sequence, compensated with the BTCOpN method, in the detected ROI-d and taking into account the values of (**D**${}_{e}$), no significant differences were found between the actual depths (**D**) of 0.2 mm and 0.4 mm or among the actual depths (**D**) of 0.6 mm, 0.8 mm, and 1.0 mm. However, between these two groups, there were statistically significant differences (see Figure 7, right).

Table 10 shows percentage values of the errors in estimated depth (**D**${}_{e}$) obtained for the ROI-d detected with the BTCOpN techniques and the BTCFN. These quantities were calculated for the defects having the greatest lateral size (**S** = 15 mm) in the actual CFRP sample. For the comparative analysis, the BTCFN method was chosen since it detected a higher number of defects compared to the remainder of the traditional techniques evaluated. The BTCFN method showed a minimum absolute error of 1.5% for **S** = 0.2 mm and a maximum of 7.0% for **S** = 0.4 mm. The BTCOpN method had a minimum value of 0.2% for **S** = 0.4 mm and 3.6% for **S** = 0.4 and 0.8 mm. In general, the error values for the two methods did not show trends in relation to the lateral sizes (**S**) of defective regions or their depths (**D**). However, the BTCOpN method in most of the depth conditions evaluated exhibited lower error values than those obtained with the BTCFN method.

The differences in the performance results could have been due to how each method accomplishes the decoupling of the background information and the information corresponding to the defects. Possibly, the optimal estimation of the parameters describing the image background favors obtaining the nonuniform heating model, in contrast to the BTCFN method, which masks the defect regions using a median filter operation.

## 4. Conclusions

This paper introduces a procedure for the automated detection of laminar defects in CFRP material inspected by pulsed thermography. For this purpose, two mutually exclusive methodologies were employed. The first uses a contrast enhancement method together with a local thresholding strategy. On the other hand, the second methodology constructs local histograms of oriented gradients with a global thresholding strategy for each thermal image.

The main contributions of our approach compared to the already known works are that our development methodologies do not require adjusting parameters or selecting reference regions and do not require prior knowledge about the state of the sample to be analyzed. Additionally, a complete or partial analysis of the temporal evolution of the temperature or its characteristics is not necessary.

Finally, our results show that on maxigrams of the actual CFRP sample containing 25 defects, the BTCOpN and BTCOp techniques correctly detected 19 and 13 flaws with 1 and 5 false detections, respectively. The performance of the BTCOpN method is superior to that obtained on the same CFRP sample recently analyzed with a deep-learning algorithm [23], where 17 correct defect detections and 4 false detections were reported. It is also superior to that obtained with the BTCFN method (18 correct detections and 1 false detection). Finally, its performance is superior to traditional contrast enhancement methods (e.g., CN, DAC, and BTCF) under the conditions evaluated. In sum-type images and raw thermal sequences, the methodology based on directional histograms of the gradient yielded AUC values greater than 0.95. This result is superior to that obtained with the G-SNR method and reported in [26], which at best only achieves AUC values with irregular trends and slightly higher than 0.85.

## Figures and Tables

**Figure 1 materials-16-02998-f001:**
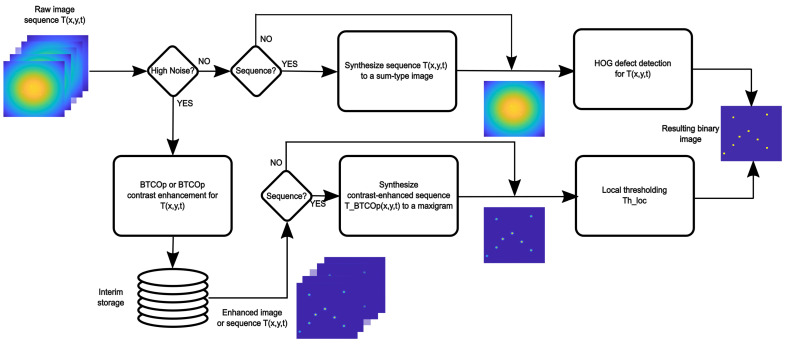
Main steps of the automated defect detection procedure.

**Figure 2 materials-16-02998-f002:**
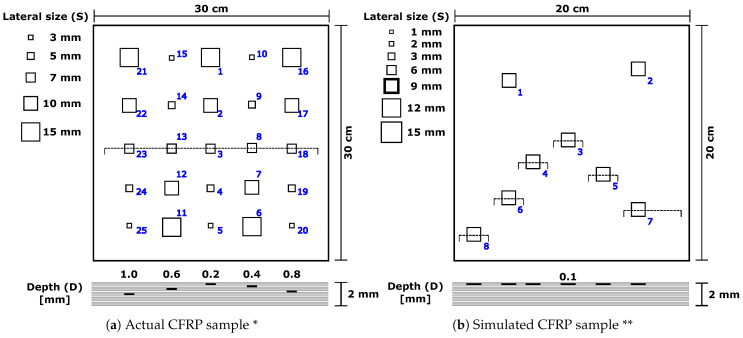
Geometry of the specimens: (**a**) actual CFRP specimen with 25 defects * and (**b**) simulated CFRP specimen with 8 defects ** (Spatial coordinates: def. 1: (40, 40), def. 2: (150, 30), def. 3: (90, 90), def. 4: (60, 110), def. 5: (120, 120), def. 6: (40, 140), def. 7: (150, 150), and def. 8: (10, 170); depths (**D**) available: 0.1,0.4,0.7,1.0,1.3,1.6, and 1.8 mm; thickness (**Th**): 0.1 mm). * Adapted from [31]. Samples and thermal images provided by the Multipolar Infrared Vision (MiViM) research group at Laval University, Quebec (Canada). ** Adapted with permission from [27] © The Optical Society.

**Figure 3 materials-16-02998-f003:**
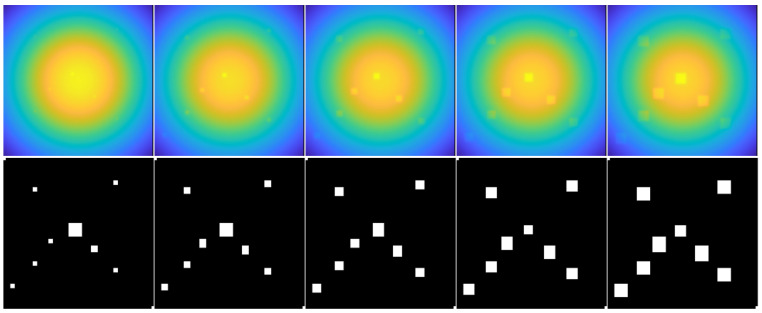
Automated defect detectionin simulated thermal sequence sum-type synthetic images. Lateral size (**S**) for ROI-d. Column No. 1: 3 mm, No. 2: 6 mm, No. 3: 9 mm, No. 4: 12 mm, and No. 5: 15 mm. Row No. 1: sum-type images, **D** = 1.0 mm. Row No. 2: resulting binary images.

**Figure 4 materials-16-02998-f004:**
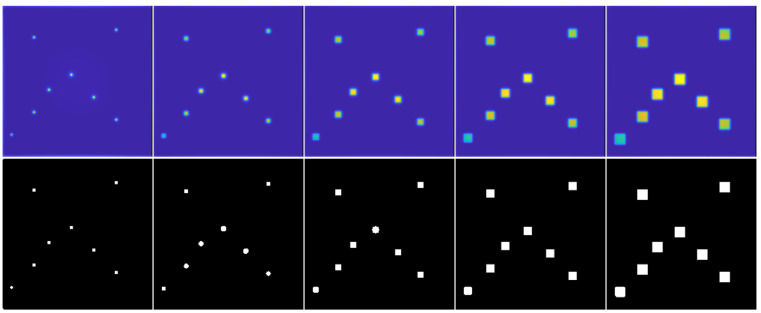
Automated defect detection in simulated thermal sequence maxigrams. Lateral size (**S**) of ROI-d. Column: No. 1: 3 mm, No. 2: 6 mm, No. 3: 9 mm, No. 4: 12 mm, and No. 5: 15 mm. Row No. 1: maxigrams, **D** =1.0 mm. Row No. 2: resulting binary images.

**Figure 5 materials-16-02998-f005:**
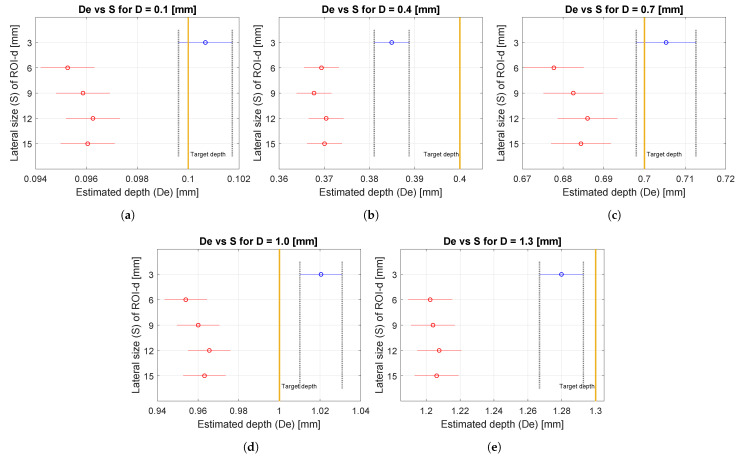
Analysis of variance (ANOVA) of estimated depth (**D**${}_{e}$) and lateral size (**S**) of defects varying the actual depth (**D**) of defects. Test performed on simulated sequence maxigrams of CFRP samples. (**a**) **D** = 0.1 mm, (**b**) **D** = 0.4 mm, (**c**) **D** = 0.7 mm, (**d**) **D** = 1.0 mm and (**e**) **D** = 1.3 mm.

**Figure 6 materials-16-02998-f006:**
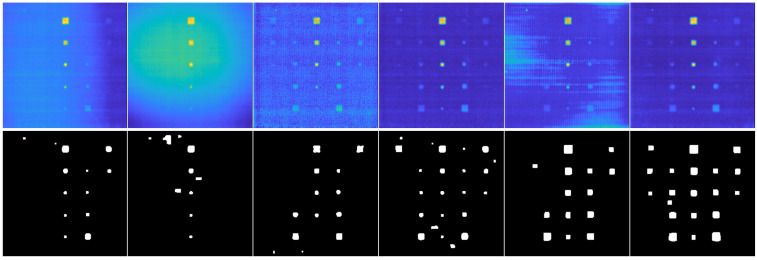
Automated defect detection in actual thermal sequence maxigrams of a CFRP sample. Row No. 1: Maxigrams of sequences contrast-enhanced with different techniques. Column: No. 1: CN, No. 2: DAC, No. 3: BTCF, No. 4: BTCFN, No. 5: BTCOp and No. 6: BTCOpN. Row No. 2: resulting binary images.

**Figure 7 materials-16-02998-f007:**
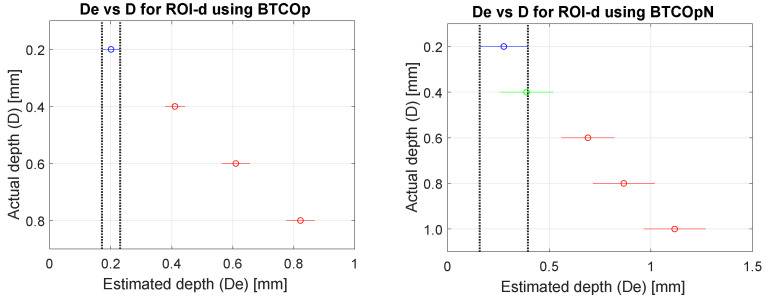
Analysis of variance (ANOVA) of estimated depth (**D**${}_{e}$) and actual depth (**D**) of defects for maxigrams of actual CFRP sequences processed using (Column 1) BTCOp method and (Column 2) BTCOpN method.

**Table 1 materials-16-02998-t001:** Acquisition conditions for the actual PT experiment. Adapted with permission from [27] © The Optical Society.

		Image	Image	Acquisition	Excitation	Acquisition	Frames
Camera	Bandwidth	Resolution	Intensity	Frequency	Source	Window	Captured
		[Pixels]	[Bits]	[Hz]		[s]	
InSb		640×512			Flash		
CCD	MWIR	Used:	14	157	photography	3.42	300
array		470×475					

**Table 2 materials-16-02998-t002:** Parameters used to adjust the simulated PT experiment. Adapted with permission from [27] © The Optical Society.

Pulse Width	Pulse Density	Acquisition Frequency	Acquisition Window
[s]	[W/m${}^{2}$]	[Hz]	[s]
$1260\times 10^{-5}$	$1\times 10^{5}$	157	9

**Table 3 materials-16-02998-t003:** Properties of the materials used.

Material	Conductivity (x = y = z Direction)	Heat Capacity	Density (ρ)
	[W/m·K]	[W·s/kg·K]	[kg/m${}^{3}$]
CFRP	0.7	1200	1600
Teflon	0.25	1050	2170
Air	0.07	928	1.3

**Table 4 materials-16-02998-t004:** AUC indicator for simulated thermal sequence sum-type synthetic images.

Depth			AUC		
(D) (mm)	S = 3 (mm)	S = 6 (mm)	S = 9 (mm)	S = 12 (mm)	S = 15 (mm)
0.1	0.99	0.99	0.98	0.98	0.97
0.4	0.99	0.99	0.98	0.97	0.97
0.7	0.97	0.99	0.98	0.97	0.97
1.0	0.96	0.97	0.97	0.96	0.97
1.3	0.95	0.97	0.97	0.97	0.97

**Table 5 materials-16-02998-t005:** AUC indicator for maxigrams of simulated thermal sequences.

Depth			AUC		
(D) (mm)	S = 3 (mm)	S = 6 (mm)	S = 9 (mm)	S = 12 (mm)	S = 15 (mm)
0.1	0.94	0.97	0.98	0.99	0.99
0.4	0.72	0.85	0.90	0.92	0.94
0.7	0.72	0.85	0.90	0.92	0.94
1.0	0.72	0.85	0.89	0.92	0.94
1.3	0.71	0.90	0.89	0.92	0.93

**Table 6 materials-16-02998-t006:** Relative percent error (%) between estimated depth (**D**${}_{e}$) and actual depth (**D**) of defects for simulated thermal sequence maxigrams.

Depth			Error [%]		
(D) (mm)	S = 3 (mm)	S = 6 (mm)	S = 9 (mm)	S = 12 (mm)	S = 15 (mm)
0.1	0.68	4.74	4.14	3.75	3.96
0.4	3.77	7.65	8.07	7.39	7.49
0.7	0.76	3.18	2.50	2.00	2.23
1.0	2.05	4.60	4.00	3.45	3.68
1.3	1.55	7.51	7.38	7.10	7.22

**Table 7 materials-16-02998-t007:** Performance indicators for maxigrams of actual thermal sequences using different processing techniques.

Method	Precision	Recall	Accuracy	F-Value
CN	0.94	0.39	0.98	0.55
DAC	0.45	0.18	0.97	0.26
BTCF	0.92	0.56	0.99	0.69
BTCFN	0.85	0.72	0.99	0.79
BTCOp	0.58	0.61	0.97	0.60
BTCOpN	0.56	0.84	0.97	0.67

**Table 8 materials-16-02998-t008:** Estimated depths (**D**${}_{e}$) and relative percentage errors (%) of defects for maxigrams of actual thermal sequences processed by the BTCOp method.

Depth		(S) [mm]				
(D) [mm]		3	5	7	10	15
0.2	**D** ${}_{e}$	0.18	0.20	0.23	0.20	0.20
	Error (%)	10.46	−0.39	−13.04	1.58	2.28
0.4	**D** ${}_{e}$	-	0.37	0.42	0.44	0.42
	Error (%)	-	8.30	−4.27	−8.86	−5.68
0.6	**D** ${}_{e}$	-	-	-	0.62	0.60
	Error (%)	-	-	-	−4.06	0.71
0.8	**D** ${}_{e}$	X	-	-	0.86	0.78
	Error (%)	X	-	-	−7.96	2.48
1.0	**D** ${}_{e}$	X	-	-	-	-
	Error (%)	X	-	-	-	-

(-) ROI-d not detected; (X) ROI-d not defined in the ground truth.

**Table 9 materials-16-02998-t009:** Estimated depths (**D**${}_{e}$) and relative percentage errors (%) of defects for maxigrams of actual thermal sequences processed by the BTCOpN method.

Depth		(S) (mm)				
(D) (mm)		3	5	7	10	15
0.2	**D** ${}_{e}$	0.52	0.24	0.19	0.23	0.20
	Error (%)	−161.26	−19.88	3.78	−13.91	−0.52
0.4	**D** ${}_{e}$	-	0.55	0.40	0.44	0.40
	Error (%)	-	−36.15	−1.07	−10.00	−0.25
0.6	**D** ${}_{e}$	-	0.94	0.69	0.59	0.62
	Error (%)	-	−55.89	−15.33	1.36	−3.65
0.8	**D** ${}_{e}$	X	-	0.94	0.94	0.77
	Error (%)	X	-	−17.86	−16.97	3.61
1.0	**D** ${}_{e}$	X	-	1.28	1.10	1.01
	Error (%)	X	-	−28.25	−9.68	−0.50

(-) ROI-d not detected; (X) ROI-d not defined in the ground truth.

**Table 10 materials-16-02998-t010:** Relative percent error (%) for ROI-d detected in maxigrams of actual thermal sequences processed by the BTCFN and BTCOpN methods (**S** = 15 mm).

	Lateral Size (S) (mm)				
Method	0.2	0.4	0.6	0.8	1.0
BTCFN *	−1.5	7.0	−2.3	−2.9	2.3
BTCOpN	−0.5	−0.2	−3.6	3.6	−0.5

* Values taken from [10].

## Data Availability

No new data were created or analyzed in this study. Data sharing is not applicable to this article.

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
