# Peer review of "Defect Detection and Depth Estimation in Composite Materials for Pulsed Thermography Images by Nonuniform Heating Correction and Oriented Gradient Information"

_materials, 2023, doi:10.3390/ma16082998_

Round 1

Reviewer 1 Report

Authors implemented defects detection and depth Estimation algorithm based on BTCOp and HOG in Pulsed Thermography Images. The article is well written, the result and the discussion is significant and interesting. However, some improvements in this paper should be made.

1) In Line 48 the authors state “According to the references consulted, few procedures have addressed the task of automatically detecting defects in thermal images as a preliminary step in characterization processes [14–18].” Please state something more about this 5 articles.

2) About Eq. 2, parameters ω and β need to be explained.

3) In Line 132 “Eq.” needs to be corrected to “Eq. 5”.

4) The definition of the AUC is ambiguous.The calculation of indicator AUC needs to be explained in detail. In addition, would it be more appropriate to use the IOU (Intersection over Union) as the indicator ?

5) If possible, the time complexity of algorithms mentioned in this paper might be compared.

Reviewer 2 Report

Dear Authors;

In general, the scientific part of this work is quite interesting and up to the standard of the journal of “materials”. However, a major revision of the current form is clearly needed, as elaborated below.

- The title is too long: it should be revised.

- The Authors should more clearly emphasis the novelty of their work in the abstract and introduction.

- I strongly recommend that the Authors add new references in the introduction.

- Fig. 1 must be placed before the section 2.1.

- A comparison with the existing published literature and discussion of the results are missing.

- Page 10; L278-L279: “However, they exhibited higher De values than those found for a 279 lateral size

of 3 mm”. Authors need to explain the reason.   

- Conclusions must be rewritten based on experimental data and theoretical data.

- Authors need to pay attention in Reference section also for formatting.

Reviewer 3 Report

The purpose of an abstract is to summarize the contents of the paper, with all the key points, but none of the details. Producing an abstract is an essential part of this process and it requires careful planning if it is to fulfil its purpose correctly. Research paper abstracts have always played a crucial role in explaining your study quickly and succinctly. It highlights key content areas, your research purpose, the relevance or importance of your work, and the main outcomes. The function of the abstract is to outline briefly all parts of the paper, not only the results. The abstract must describe what you have studied in your research and what you have found and what you argue in your paper. And never include numerical results in your abstract. You have plenty of time to present and refer to these in the body of your paper.

The aims and objectives of the research are well defined.

The structure of paper is good, and the additions bring improvements. The basic structure of a typical research paper is the sequence of Introduction, Methods, Results, and Discussion. The paper is well–structured and its parts are logically interconnected. Overall, this manuscript is well–written and interesting to read.

The INTRODUCTION section provide the necessary background information and is an extensive review of the literature. The INTRODUCTION is quite succinctly presented, but on the whole, it is comprehensive with the state of research in the field, although important discussions are missing and there is not much explanation on the research carried out. 

The METHODS section describe the context and setting of the study. This section provide the readers with sufficient detail about the study methods to be able to reproduce the study if so desired. Thus, this section is relative specific, concrete, technical, and fairly detailed.

The body of paper describe the important RESULTS of the research and present key findings. The purpose of a RESULTS section is to present the key results of your research. It is important to plan this section carefully as it may contain a large amount of scientific data that needs to be presented in a clear and concise fashion. There is far too much information, the presentation is wide and bushy. I recommend presenting these results in several articles, always focusing on distinct results series. This part occupies a fairly important space within the article. Although the synthesis is good, it is a little too long presented.

The CONCLUSION section succinctly summarize the major points of the paper and is quite succinctly presented. Are only technical comments, very briefly presented, without other opinions, conclusions or remarks. I would recommend presenting the novelties of this study, the main characteristics and the major conclusion that individualize this research. Indicate what you have done that is new compared to the already known works.

FIGURES are particularly important because they show the most objective support of the research. The TABLES are representative and the FIGURES & GRAPHS have good qualities.

The list of REFERENCES is long and relatively well chosen. The entire bibliography is current, and modern works are mainly used. Literature review provides comprehensive information about the current state of research.

Reviewer 4 Report

see attachment

Round 2

Reviewer 2 Report

Authors have responsed carefully to my all comments. Manuscript may be accepted for publication.